# Breathing Rate as a Marker for Noise-Induced Stress in Guinea Pigs

**DOI:** 10.3390/brainsci15111152

**Published:** 2025-10-27

**Authors:** Mark N. Wallace, Joel I. Berger, Christian J. Sumner, Alan R. Palmer, Michael A. Akeroyd, Peter A. McNaughton

**Affiliations:** 1Hearing Sciences, School of Medicine, University of Nottingham, Nottingham NG7 2RD, UK; alan.palmer@nottingham.ac.uk (A.R.P.); michael.akeroyd@nottingham.ac.uk (M.A.A.); 2Human Brain Research Laboratory, Department of Neurosurgery, University of Iowa Hospitals and Clinics, Iowa City, IA 52242, USA; joel-berger@uiowa.edu; 3Auditory Neuroscience, Department of Psychology, Nottingham Trent University, Nottingham NG1 4FQ, UK; christian.sumner@ntu.ac.uk; 4Wolfson Sensory, Pain and Regeneration Centre, King’s College London, London WC2R 2LS, UK; peter.mcnaughton@kcl.ac.uk

**Keywords:** motion tracking, sniffing, tinnitus, gap prepulse inhibition of the acoustic startle, startle reflex, mice, inspiration

## Abstract

**Background:** Breathing rate is affected by physical stressors such as temperature or hypercapnia and by psychosocial stressors such as noise or overcrowding. In behavioral tests for tinnitus, rodents are often presented with trains of startle pulses. We postulated that using these pulses at successively higher sound levels would produce a cumulative increase in stress. Here, we demonstrate a novel means of assessing that increase in stress. **Methods**: By placing pairs of reflective markers on the abdomen and using a motion tracking system, we were able to remotely measure respiratory movements. A series of 20 startle pulses were presented in sequence at seven increasing sound levels, and changes in respiratory rate were tested with the Wilcoxon matched-pairs signed rank test and the Friedman Test. **Results**: Markers placed on 20 alert active mice showed evidence of sniffing behavior but no purely respiratory signal. By contrast, in all 18 guinea pigs, abdominal markers did track respiratory movements. The breathing rate in guinea pigs was similar to previous studies: (mean 104 ± 13; range 86–131 bpm). Presenting quiet startle pulses to guinea pigs caused a significant increase in breathing rate (by about 20%), even with pulses at 75–80 dB SPL. Increasing pulse sound levels in the range of 85–105 dB SPL did not reliably produce any further increase in breathing rate. **Conclusions**: We propose that tracking abdominal movement may allow measurement of psychosocial stress in the guinea pig. Once an animal is startled, increasing the pulse sound level did not produce any further increase in stress levels.

## 1. Introduction

Animals are often used to study changes produced in the body by either acute or chronic stress. Tinnitus is one such condition associated with or exacerbated by stress. Indeed, recent studies in the rat have shown evidence that tinnitus can be produced by physical (restraint) stress alone, without any direct damage to the auditory system [1,2]. This ties in with epidemiological studies, which have shown that some people with tinnitus do not have any significant hearing loss, while many people with moderate hearing loss do not have tinnitus [3]. The presence of psychosocial stress has the same predictive value for developing tinnitus as occupational noise [4,5], with stress generally accounting for about 19% of tinnitus risk and noise exposure for 27% [6]. The distress caused by tinnitus in humans is linked to stress levels [5,7,8,9,10], and the demand for an effective treatment for tinnitus is especially strong among patients with high stress levels [11]. Thus, it would be useful to be able to measure the stress levels in animal models of tinnitus [8,12,13]. Increased levels of stress have already been shown by behavioral testing in rats, where tinnitus had been induced either by salicylate or noise exposure [14], and dual testing of tinnitus and stress may become more frequent in future.

There have been many attempts to develop drug treatments for tinnitus, and initially, these are tested on animal models. However, it is difficult to extrapolate from rodents to humans, and in the past, drugs that have seemed effective in a rodent model of tinnitus turned out to be ineffective in the clinic [15]. Since stress is a factor in tinnitus, its assessment in animal models of tinnitus may be important when developing therapeutic interventions. Drugs that can reduce both the stress levels and the changes in neural plasticity that are thought to underlie tinnitus [16,17] may be more effective than drugs that only target one aspect of tinnitus. Stress is difficult to measure directly in animals because of the complexity of the systems controlling the response [18]. There are two main regulatory systems for responding to stress: the hypothalamic–pituitary–adrenocortical (HPA) axis, which leads to the release of glucocorticoid hormones, such as corticosterone, from the rodent adrenal cortex, and the sympathetic nervous system, which involves a network of noradrenergic neurons and the release of adrenalin from the adrenal medulla. The sympathetic system directly innervates respiratory centers in the brainstem to modulate the respiratory rate. Corticosterone metabolites can be measured in blood or salivary samples or non-invasively through the collection of fecal pellets. None of these procedures are ideal, because blood and saliva removal are potentially stressful procedures [19,20], and there is a delay between the release of corticosterone by the adrenal cortex and the production of the fecal pellets of about 8 h in rats [21]. Changes in stress levels can be more immediately assessed by monitoring the breathing rate and heart rate [9,22]. The breathing rate has been measured in guinea pigs by using a whole-body plethysmograph or intrathecal canulae, and the original studies [23,24,25] established the guinea pig as a useful model for studying mammalian respiration.

Many more methods for measuring the breathing rate in rodents are now available [26,27], but these often involve restraint, intranasal surgery, and tethering or have other drawbacks. We wanted to avoid any physical restraint in our animals, as it is potentially distressing [28], and we hoped to be able to measure the breathing rate more simply by placing surface markers on the thorax or abdomen. These markers can be easily tracked using infrared cameras and a motion tracking system [29], which we currently use to measure the acoustic startle response in the gap prepulse inhibition of the acoustic startle (GPIAS) method for identifying tinnitus [30,31]. The two animal species we have been employing as models of tinnitus are guinea pigs and mice, and we sought to measure the breathing rate in both these species by tracking the movement of markers placed on the dorsal thorax or abdomen before and during the presentation of startle pulses used to identify tinnitus.

The first objective of this study was to determine whether reliable measurements of the breathing rate could be made by placing reflective markers on the abdomen of freely-behaving animals. The second and subsidiary objective was to determine whether the presence of trains of startle pulses would produce an increase in the respiratory rate. The third objective was to determine whether there was a linear relationship between the breathing rate and the sound level of the trains of startle pulses.

The GPIAS method involves presenting trains of startle pulses, and this testing would be expected to increase the stress levels, especially in guinea pigs, who are adversely affected by being startled [32]. During the initial testing phase, a series of startle pulses were presented at different sound levels of between 70 and 105 dB SPL, in order to determine the optimal sound level for identifying GPIAS [33]. We have been concerned that presenting these startle pulses may increase stress levels and therefore alter the perception of the tinnitus and that this effect would be more pronounced at the highest sound levels. We therefore assessed the effect of startle trains on the stress levels of animals and determined whether higher sound levels of the startle pulses were associated with higher levels of stress. In practice, we were unable to record a stable breathing pattern in the mice, which would have provided a suitable baseline for comparison with the experimental conditions. By contrast, after a few minutes in the recording chamber, the guinea pigs showed stable breathing patterns, within the normal range for resting animals, The breathing rate increased significantly when the animals were startled, but once startled, increasing the sound level of the acoustic pulses did not produce any further consistent increase in breathing rate.

## 2. Materials and Methods

### 2.1. Animals

Male Dunkin–Hartley guinea pigs (*n* = 18) were purchased from a registered breeder (Charles River, Margate, Kent, UK) at a weight of 300–350 g and were allowed to acclimatize in groups of 10 in a floor pen containing disposable cardboard shelters, sawdust, and bedding material. The temperature in the room was controlled to keep it in the range of 15–21 °C, and the humidity was kept at 45–65%. Clear differences exist in the responses of male and female animals to restraint stress [34], but here we only used male animals to have smaller more coherent groups. For the sake of consistency, we avoided using female mice as well. A total of 20 C57BL/6J male mice, aged 3 months were supplied by Charles River. All animals were allowed to recover from the journey for at least a week before any testing was started. Mice were group-housed (*n* of 2–4) in individually ventilated cages on a 12:12 h light: dark cycle with food and water available ad libitum and with the temperature and humidity kept at 20–24 °C and 40–70%.

### 2.2. Measuring Movement

We used motion-tracking software that was designed mainly for use with humans [29] but which we have adapted for use with guinea pigs and mice [31] by the use of small cameras with a short focal length. They consisted of three infrared OptiTrack Flex 3 cameras (Natural Point, Corvallis, OR, USA), focused on reflective 4 mm wide self-adhesive markers. These cameras were placed in a triangular pattern 300 mm apart at 750 mm above the animal within a small audiology booth covered with foam wedges. The booth contained a window, which was partially blocked by foam wedges, so that light could enter at the top, but animals were not distracted by any movement in the laboratory. The guinea pig set up had an open-topped wire cage with non-reflective paint (310 mm long by 155 mm wide and 155 mm tall). Even large guinea pigs could turn around in this cage and occasionally they did climb out. A loudspeaker was placed just above the left end of the guinea pig cage. Mice were placed in a two-liter polyethylene bowl with the loudspeaker directly above it. The bowl contained a layer of sawdust with seeds scattered among it. The sawdust was changed each day but not between testing mice housed in the same cage, so that it would smell like their home cage and help minimize stress. The chamber was cleaned with mild disinfectant between each use.

Prior to each use, the camera system was optimized using a dummy with reflective markers to optimize the LED (light-emitting diode) levels and the filter settings to reduce the background signal. Software filters were set to only accept a signal from a small round source. The system was calibrated at least once a week to ensure optimal performance where the residual mean error, calculated by triangulation, was less than 0.1 mm. Small (3–4 mm) reflective markers with self-adhesive backing (MCP1130, Natural Point, Corvalis, OR, USA, NaturalPoint.com) or circles of reflective tape were placed on the fur. For guinea pigs, pairs of markers were placed on the dorsal abdomen or thorax and on a cartilaginous ridge at the dorsal edge of the pinna to monitor ear flicks. The guinea pigs were unconcerned by the presence of the markers and almost never tried to remove them during the recording period. By contrast, mice found the markers irritating and usually tried to remove them immediately. Mice were held by their tails and their hind legs lifted while self-adhesive markers were applied to different points on their body. They sometimes allowed markers to remain on their body but generally removed them during grooming within a few minutes of placement. The mice tolerated the 3 mm markers better than the 4 mm markers, but the cameras provided an incomplete recording trace with the smaller markers, and there were breaks in the recording. This was also true of the two-dimensional tape. Thus, we only report the results with the 4 mm markers here. We tried to place the markers on the mice as quickly as possible and with minimal restraint, as even holding by the tail may increase stress levels [35]. To minimize stress, mice were moved between locations in a cardboard tube or with cupped hands.

The cameras recorded the movement of the markers with a 100 Hz sampling rate controlled by the OptiTrack Motive 2 system. The motion-tracking software triangulated the absolute position of the markers and produced three columns of data for each marker corresponding to the *x*, *y*, and *z* coordinates. The data were exported as .csv files and analyzed in an Excel spreadsheet or in Matlab^®^ (R2009b, MathWorks, Natick, MA, USA). The Euclidean distance between the markers was then calculated to provide a measure of relative distance and to filter out the movements of the mouse within the cage.

### 2.3. Acoustic Stimulation

Sound files with narrow or broadband background noise and variable amplitudes for the startle pulses were synthesized as 16-bit digital waveform files (.wav) using Adobe Audition 1.5 (Adobe Systems Incorporated, San Jose, CA, USA). Each file contained a series of 20 startle pulses, 10 of which had a preceding gap, in a pseudorandom order with intervals of 6 s on a continuous background of narrow band noise (filtered at 4–6 or 16–18 kHz) or broadband noise (BBN). For each animal, we recorded the responses to at least one set of pulse trains, and in eight of them, a sequence of seven pulse trains was presented in ascending order with pulses at 75 dB up to 105 dB in 5 dB steps against a background noise at 70 dB SPL. Animals were never kept in the recording chamber for more than an hour at a time, and no more than two recording sessions were made in one week. The 50 ms long gaps (2 ms rise/fall time) ended 50 ms before the startle pulses (20 ms long with 1 ms rise/fall times), composed of BBN (50 Hz to 20 kHz). Digital to analogue conversion was made with a Tascam US-144 interface (TEAC Professional Division, Santa Fe Springs, CA, USA) and taken via an Onkyo sound amplifier to a single 25 mm loudspeaker (Tymphany Peerless Gold XT25BG60-04 tweeter, Falcon Acoustics, Stanton St John, Oxford, UK). Some guinea pigs stayed almost motionless for up to 50 min during the sequence of trials, but others were more active. All the mice were active during the trials and were almost continuously moving around or digging in the sawdust. These animals were being used for a study of tinnitus as described previously [31]. Each trial of 20 startle pulses lasted for 120 s, and each session consisted of 7–14 trials.

### 2.4. Statistical Analysis

Breathing rate in the “silence” condition was measured at the end of a recording session when the animal was acclimatized to being in the recording chamber. At least 4 min. was left after the end of any startle stimulation before recording. The mean breathing rate, for all 8 guinea pigs in silence, was then compared to the mean breathing rate for the same animals in the 7 conditions, where startle pulses were presented in sequence from 75 dB to 105 dB SPL, using the Friedman Test. This allowed the null hypothesis to be tested—that there was no significant difference between silence and any of the stimulus conditions. The Friedman Test was also used to compare the 7 stimulus conditions, where the null hypothesis was that there was no significant difference between any of the stimulus conditions. The range of values, used to measure the mean of each condition, was not normally distributed, and a non-parametric test (Wilcoxon matched-pairs signed rank test) was used to compare the silence condition to the first startle condition (75 dB pulses).

A regular breathing cycle at rest means that the breathing pattern can be modeled by one or more sine waves and analyzed by a fast Fourier transform (FFT) to separate out different frequency components within the signal. Two-minute-long segments of breathing in silence were analyzed in this way, and if there was a simple regular breathing rate, then this produced a single large peak indicating the dominant frequency. If the breathing rate was less stable, then multiple frequencies would be present, and this would be indicated by multiple peaks where no one frequency was dominant.

The regular saw-tooth nature of the abdominal movements meant that it was possible to measure the proportion of time the abdomen was expanding/contracting, and this roughly corresponds to inspiration/expiration, with an upwards inflection in the marker reflecting the expansion of the abdomen (inspiration) and a downwards inflection reflecting the contraction (expiration). The proportion of time spent in inspiration/expiration was extracted from the two-minute-long periods of breathing at rest and the larger number (expiration) plotted against the breathing rate for all 18 guinea pigs. The linear regression was then calculated to determine whether there was a correlation between the two variables. If an increased breathing rate caused a reduction in the time spent in the expiratory phase, then there would be a strong correlation. The strength of the correlation was indicated by the square of the regression value (R^2^), where a value close to 1 indicates a very strong correlation, and a value close to 0 indicates that the breathing rate by itself has little predictive power over the time spent in expiration.

## 3. Results

### 3.1. Breathing Patterns in C57BL/6J Mice

C57BL/6J mice are usually very active and inquisitive. We observed that when placed in a new environment they tended to explore and were seldom stationary. They did not like foreign objects on their fur and either tried to shake or groom them off within a few minutes of them being attached. Despite this, the skin over the abdomen is less sensitive than that over the head region, and some mice were prepared to keep a pair of markers on the abdomen for a few minutes at a time. We plotted the distance between the two markers in an attempt to record abdominal movements that would indicate the breathing pattern but were unable to do so. The distance between the markers varied as the mouse moved around the chamber or scratched in the sawdust, but there was no sign of a regular breathing pattern in the expected range of 2–4 Hz (Figure 1a). There were regular small abdominal movements, but these were in the range of 7–11 Hz and were of low amplitude (see small arrows in Figure 1b). These corresponded to sniffing behavior as the mouse sampled the air in the chamber and is a typical behavior for mice placed in a new environment with different smells.

While the mice were in the chamber we presented trains of 20 startle pulses to them, with a period of 6 s between each pulse. At the higher sound levels of 90–105 dB SPL there were clear abdominal responses that corresponded to the muscular contractions produced by the startle response. Two examples of the abdominal response to a sound pulse at 105 dB SPL are indicated by the large arrows in Figure 1b. Thus, the tracking cameras are clearly able to detect small abdominal movements in the mouse, but for this strain at least, while the mouse was active, we could not detect regular abdominal movements linked purely to respiration.

### 3.2. Breathing Patterns in Guinea Pigs at Rest

Relaxed guinea pigs show a regular saw-tooth pattern of abdominal movements that reflect the breathing pattern (Figure 2). However, even the act of removing the guinea pig from its floor pen and placing it in the recording chamber is stressful. This stress is compounded by the fact that the animals are isolated. The degree of handling and isolation stress seems to vary widely with some animals appearing relaxed and others appearing agitated. This is reflected in the breathing pattern with some animals having slow regular patterns (Figure 2c), while others have a more rapid and irregular pattern. Thus, even within a resting period, there can be variation in the respiratory rate with short bouts of a higher rate (marked by horizontal bars in Figure 2e). There can also be short bouts of sniffing (marked by horizontal bars in Figure 2g). One of these sniffing periods is shown on a faster time scale in Figure 2h. These sniffing periods are characterized by 6–11 rapid amplitude fluctuations at a rate of 7–11 Hz. Measurements of the mean breathing rate at rest were made for 18 guinea pigs in two groups. The mean was 104.2 (±13.25) with a range of 86–131 bpm.

The proportion of time spent in inspiration varied from breath to breath, but the mean values also varied between animals. An example of a typical relaxed breathing cycle with similar times spent on inspiration and expiration is shown in Figure 2d. In some breaths, there was a more rapid inspiration phase followed by a longer expiration phase (Figure 2f). The mean proportion of time spent in expiration (or the inverse for inspiration) was measured over two minutes of rest for the 18 guinea pigs. The mean percentage of time in expiration was 55.7 (±4.36)%, with a range of 49–64%. The mean expiration value at rest was plotted against the mean breathing rate, but there was only a very weak correlation (Figure 3). The breathing trace was also analyzed with an FFT and when there was a regular breathing pattern the main peak in the FFT did correspond to the breathing rate (Figure 2a,b). However, when the pattern was more irregular, the FFT had multiple peaks with similar amplitude that made it difficult to identify a dominant frequency for the breathing rate.

The guinea pigs were further stressed by presenting them with trains of 20 startle pulses, half with a preceding gap in the background noise, of the type used in baseline testing before initiating a tinnitus-producing procedure. The presence of a startle pulse of more than 75 dB SPL produced a sharp intake of breath (expansion of abdomen), which was followed by an equally sharp expiration. At the higher sound levels tested, this sharp intake of breath was often followed by a larger than usual inspiration as part of the regular breathing cycle. Otherwise, there was little disruption of the regular breathing pattern. This is illustrated in Figure 4, where the regular breathing pattern at rest (Figure 4a) is altered by the presentation of a train of startle pulses (105 dB) superimposed on a background of noise at 70 dB SPL (Figure 4c). The presence of a startle pulse produced a rapid twitch in the abdominal muscles, which was reflected in a rapid increase and decrease in the distance between the abdominal markers (small vertical arrows in Figure 4b). As expected, this twitch was smaller when preceded by a 50 ms gap in the background noise (oblique small arrows in Figure 4b). In each case, the twitch was followed by a larger than usual amplitude in the next cycle of breathing. Apart from this, there was only a small increase in the number of irregular breathing cycles and a small increase in the breathing rate (from 97 to 103 bpm). An expanded view of the alteration in the breathing pattern produced by a startle pulse is shown in Figure 4d.

The trains of GPIAS pulses were presented to each animal in a series with 5 dB steps in the pulse sound level and a constant 70 dB SPL noise background in the range of 75 to 105 dB. When the amplitude of the brief abdominal twitch was measured across the population of guinea pigs there was a smooth sigmoidal increase in the amplitude as shown in Figure 5.

Our main objective was to determine whether or not there was a similar sigmoid increase in breathing rate when trains of startle pulses were presented over the corresponding set of sound levels. Breathing rate was measured during the presentation of a series of pulse trains at different sound levels from 75–105 dB SPL against a background of narrowband noise (4–6 kHz) at 70 dB SPL and compared to the breathing rate during a period of silence in 8 guinea pigs. The mean breathing rate in silence was 102 (±10.9) bpm, and this increased to values of 114–120 bpm during periods of stimulation with startle pulses (Figure 6). When the breathing rate values were compared across all 8 groups for each sound level, including silence, using the Friedman Test there was a highly significant difference (*p* = 0.005, Friedman statistic 20.1). However, when the results for the 7 groups stimulated between 75–105 dB were compared there was no significant difference found by the Friedman Test (*p* = 0.49, Friedman statistic 5.5). When the Wilcoxon matched-pairs signed rank test was used to compare the mean values for silence and the startle pulses at 75 dB there was a highly significant difference (two-tailed *p* = 0.008). Thus, the results show that startling the guinea pig does produce a significant increase in breathing rate, but once the guinea pig is startled, increasing the sound level of the startle pulse does not produce any further increase in breathing rate.

## 4. Discussion

We have shown that measuring abdominal contractions using a motion tracking system in guinea pigs allowed us to successfully track their breathing rate and subsequent stress related to startling sounds. The system’s main advantage is that it is a non-contact method that avoids any surgical implantation, tethering, or restraint in a confined space. By contrast, we were unable to measure a breathing rate for active mice, as we were unable to show any regular movements purely linked to respiration. The main airflow movements we detected in mice were regular sniffing movements at 7–11 Hz. Regular sniffing movements are typical of mice placed in a new environment as they actively sample for novel odors [36,37]. These sniffing responses may also have been induced by the startle pulses, as startle pulses in rats can induce bouts of sniffing behavior [38], and that is also true in mice [36]. There were also movements associated with abdominal contractions as the mouse actively explored the recording chamber, but we were unable to detect the regular saw-tooth style breathing movements at 2–4 Hz typical of a mouse at rest [39]. Mice alternate between periods of quiescence where their respiratory rate can drop as low as 150 bpm to more active periods of exploration when their respiratory rate is at least twice this [40] and may be as high as 7 Hz [41]. During periods of active exploration, the needs of gaseous exchange may all be met by sniffing, where the rate increases to a range of 8–12 Hz [36,37]. Rodents such as mice, rats, and guinea pigs are obligate nose breathers [26], and thus the periods of sniffing serve dual functions of gaseous exchange as well as odor discrimination. More sophisticated methods of measuring mouse respiration are now available, such as nasal thermistors to detect temperature differences of the tidal airflows [39], direct placement of canulae in the nasal cavity [36] or thorax [37], or continuous high resolution video monitoring in the home cage [19], and these have supplemented the more traditional whole-body plethysmography [28]. However, all of these methods involve some degree of restraint, tethering, or attachment of telemetry devices [37] or keeping the mouse in its home cage and are not suitable for making measurements during tinnitus testing using startle pulses. Mild alerting stimuli, such as a quiet click, increase the breathing rate in mice, without any increase in heart rate. This is caused by an increase in the proportion of time spent sniffing and may not be an indicator of increased stress [42]. Thus, we concluded that measuring abdominal movement during tinnitus testing in active mice is not an appropriate method for monitoring changes in respiratory rate or stress levels.

The situation with guinea pigs is very different, as they are generally much more placid and less active than mice. Mice have a much higher metabolic rate, linked to their small body size which is about 4% that of a guinea pig. In the wild, they tend to be constantly on the move searching for highly nutritious foods such as seeds, nuts, fruit, or small animals, and mice are constantly sniffing the air to detect changes that may indicate danger or the presence of food. This characteristically high level of activity is still present in the laboratory setting [39]. By contrast, guinea pigs are hystricomorph rodents with an herbivorous lifestyle involving chewing large quantities of less nutritious vegetable matter such as hay or cabbage [32]. They are also very gregarious animals and isolating them by placing them in a strange environment, such as the recording chamber, will often induce periods of immobility or at least reduced exploratory activity. This meant we were able to record periods of regular saw-tooth style breathing in all our guinea pigs. Brief bouts of sniffing were occasionally superimposed on this, but they had a sharp onset, and their relatively high rate (about 8 Hz) meant they could be easily identified and removed when calculating the respiratory rate at rest.

The mean value for rest breathing recorded for our guinea pigs was 104 ± 13 bpm, and this is in the middle range for previous studies made with the traditional plethysmograph. In Guyton’s [23] classic study, the adult guinea pigs were well acclimatized to the plethysmograph and gave a mean rate of 90 bpm with a range of 69–104, while Nixon [25] gave rest values of 131 ± 14 bpm. This is a large difference, but in Nixon’s [25] study, the effect of physical exercise was being measured, and the higher resting levels may have been an anticipatory response to having to run up and down a slope. The study by Amdur and Mead [24] only used infant guinea pigs, and their mean value of 84 ± 14 bpm may have been lower because of techniques used to reduce the stress associated with keeping them in the plethysmograph chamber for up to 5 h or the after-effects of the ether anesthesia.

There are many different types of stressors and a complicated set of different brain systems that are involved in the response to stress [43]. Stressors are broken down into two types: physical and psychosocial [8,44]. Physical stressors such as exercise or the presence of actively growing tumors involve an increased metabolic load and lead to an increased respiratory rate [25,27], while psychosocial stressors may not involve any significant increases in metabolic load but rather an increased mental load related to an actual or anticipated threat [43,45,46]. Both types of stressors can produce an increase in respiratory rate, but the mechanisms involved are different [47,48]. Trains of startle pulses such as those often used in tinnitus testing in rodents fall into the psychosocial category, but there does not seem to have been any previous study of their effect on the respiratory rate. Other types of psychosocial stress in rodents, such as anxiety [22], social defeat response [47], and cardiac defense response to arousal [44], all produce increases in respiratory rate, and it would be expected that the startle response would too, especially as it involves an increased level of muscle twitches. Thus, the effect of trains of startle pulses producing an increase in respiratory rate was in line with expectations. What was more surprising was the lack of any further increase in respiratory rate as the stimulus level increased. Previous studies have shown a sigmoidal increase in startle amplitude with increasing startle sound level for both the whole-body startle in mice [49] and the ear flick in guinea pigs [31]. In our data, the presence of a loud startle (>90 dB) always initiated an inspiratory cycle, and this may involve direct pathways to motor nuclei within the brainstem [50]. However, the effect on the long-term respiratory rate controlled by the pre-Botzinger complex of the pons [51] may involve a separate stress related pathway that has an all-or-none component and is not changed by the level of the startle sound.

The wide range of guinea pig respiratory rates observed both within an individual study such as ours and between different studies [23,25] suggests that the breathing rate at rest may be an indicator of the levels of chronic stress. Human psychological studies have shown that a lower respiratory rate is a correlate of higher psychological well-being and that reducing stress levels can lead to a lower respiratory rate at rest [52,53]. Similarly, in rats selectively bred for high anxiety, the breathing rate at rest was significantly higher than in rats bred for low anxiety [22]. Chronic stress is thought to be an important factor in the development of tinnitus [5,6], and once tinnitus has become established, the presence of anxiety or chronic stress is associated with higher tinnitus severity [54,55,56]. Measuring anxiety levels in rodents with the open-field or hole-board tests may be flawed because of problems with interpretation [57], and there is a need for other tests of chronic stress or anxiety to supplement those used previously [14]. A contactless method for measuring the breathing rate may provide a simple alternative. Research is continuing to identify drug treatments that would be therapeutic in some forms of tinnitus both in the clinic [58] and in rodent models [59]. Having an animal model where both tinnitus and stress levels can be measured easily and simultaneously may be a useful tool in the future testing of drugs aimed at reducing tinnitus annoyance.

## 5. Conclusions

Placing a pair of reflective markers on the dorsal abdomen of a guinea pig and tracking their motion relative to each other provides a contactless and non-invasive method for measuring the respiratory rate and the proportion of time spent in inspiration/expiration. Changes in the respiratory rate may provide a rapid (within a few minutes) indication of a change in their psychosocial stress levels. Guinea pigs are easily startled, and even quiet sounds like short (20 ms) sound pulses at 75 dB SPL are sufficient to stress them and significantly increase their respiratory rate. Increasing the sound level of the startle pulse within the range of 75–105 dB SPL did not seem to have any effect on the degree of stress produced. By contrast, abdominal markers placed on active mice did not appear to be a suitable method for monitoring psychosocial stress. The motion-tracking system could detect small respiratory movements in the form of bouts of high-frequency sniffing, but the active mice did not show a regular breathing pattern linked solely to gaseous exchange. Having a convenient way of measuring psychosocial stress, at the same time as using the GPIAS method for detecting the presence of tinnitus in guinea pigs, may provide a method for linking tinnitus and stress in the future.

## Figures and Tables

**Figure 1 brainsci-15-01152-f001:**
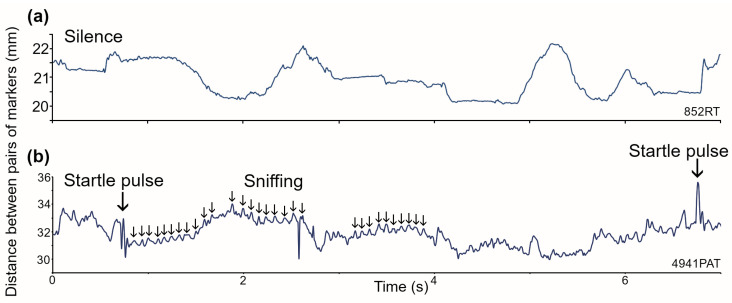
Recordings of respiratory movements in a mouse, made by calculating the distance between two reflective markers sitting on each side of the dorsal abdomen. (**a**) In the awake mouse actively moving around the recording chamber, there are changes in the shape of the abdomen but usually no sign of a regular pattern corresponding to respiration. (**b**) When a mouse is presented with trains of startle pulses, these often trigger bouts of sniffing activity that produce small regular movements of the abdomen (small arrows). When the startle pulses are presented at a high level (at 105 dB SPL here), there are sudden twitches of the abdominal muscles that produce brief abdominal movements (large arrows).

**Figure 2 brainsci-15-01152-f002:**
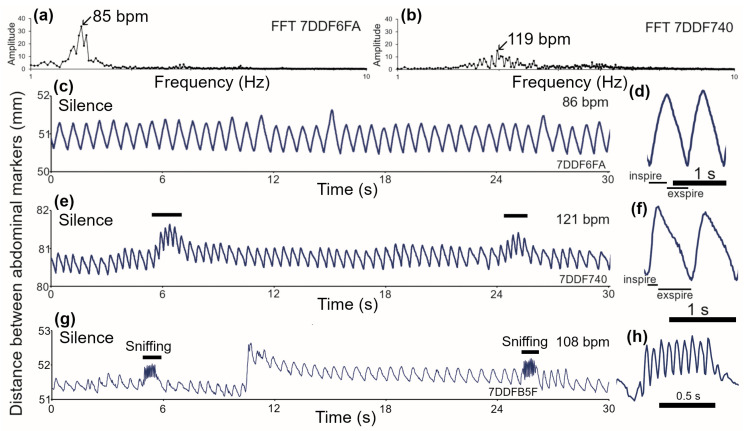
In guinea pigs, when markers were placed on either side of the abdomen, there were always regular abdominal movements that produced a saw-tooth pattern. The frequency and regularity of these movements were measured by performing fast Fourier transforms (FFTs). The first FFT (**a**) is of the slow regular breathing pattern (86 bpm) shown for animal 7DDF6FA in panel (**c**). There was a prominent peak in the FFT at 85 bpm along with two nearby flanking peaks. However, when the breathing was more irregular, as for the animal 7DDF740 (**b**), the FFT had multiple peaks, and the central peak at 119 bpm was less prominent, although still close to the number of breaths per minute (121 bpm), as shown in panel (**e**), which shows irregular short periods where the breathing speeds up (horizontal bars). When placed in a new environment, guinea pigs also show brief periods of sniffing (**g**), where there are rapid amplitude fluctuations (horizontal bars) as shown for animal 7DDFB5F, who had an intermediate breathing rate (108 bpm). Magnified views of typical breathing movements are shown for slow regular breathing, with approximately equal periods of inspiration and expiration in 7DDF6FA (**d**), more rapid breathing with a rapid inspiration phase and slower expiration phase for 7DDF740 (**f**), and a period of sniffing in 7DDFB5F (**h**).

**Figure 3 brainsci-15-01152-f003:**
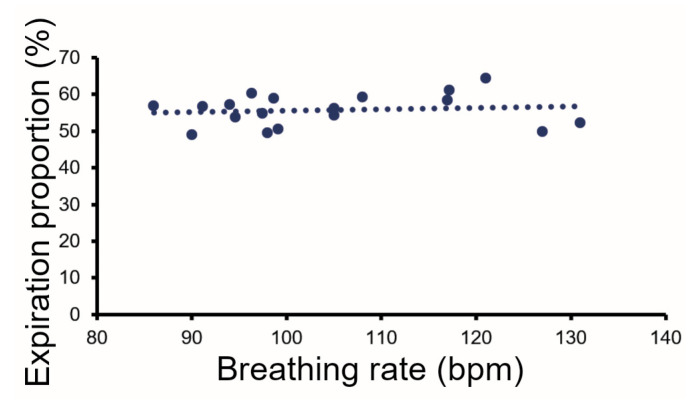
The mean period of time spent in the expiratory phase, for all 18 guinea pigs, was plotted against their mean breathing rate at rest in the chamber, but there was only a very weak correlation (R^2^ = 0.01), and the proportion of time spent in the expiratory phase remained relatively constant at different breathing rates.

**Figure 4 brainsci-15-01152-f004:**
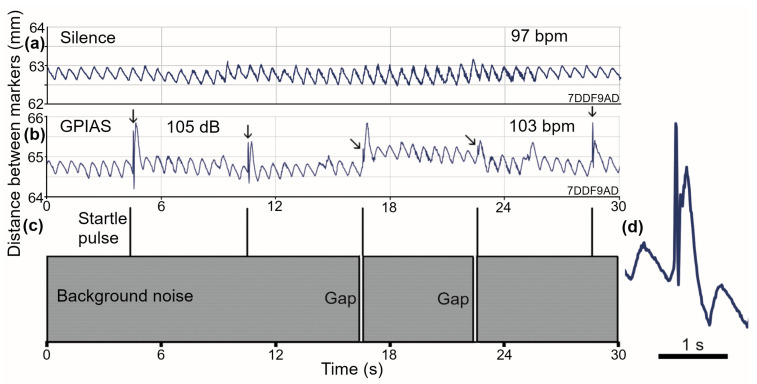
All animals were presented with the gap prepulse inhibition of the acoustic startle (GPIAS) stimuli as trains of startle pulses, where half of the pulses were pseudo-randomly preceded by a 50 ms gap in the background noise, as shown schematically for part of one sequence in panel (**c**). In the animal illustrated (7DDF9AD), there is a regular breathing rate in silence of 97 bpm (**a**), and this increased by about 6% during the presentation of the train of startle pulses at 105 dB (**b**). Each sound pulse produced a rapid and brief increase and decrease in the width of the abdomen (vertical arrows), but these twitches were reduced in amplitude when the pulse was preceded by a gap (oblique arrows). Despite this, all of the respiratory cycles that came immediately after the pulse were increased in amplitude. After this, they all settled back into their usual pattern. Three breathing cycles, centered on a pulse response, are shown on an expanded timescale in (**d**).

**Figure 5 brainsci-15-01152-f005:**
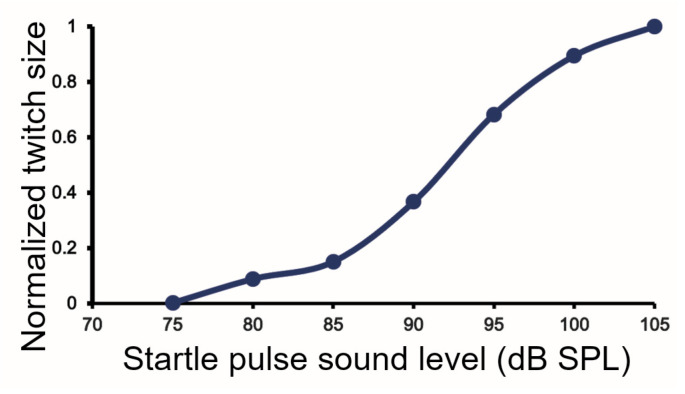
The amplitude of the brief twitch in abdominal muscles produced by the startle pulses was measured across the population of 18 guinea pigs and the size of the movement normalized against the largest movement which was produced by pulses at 105 dB SPL. The increase in amplitude formed a rough sigmoid curve when plotted against increasing sound level.

**Figure 6 brainsci-15-01152-f006:**
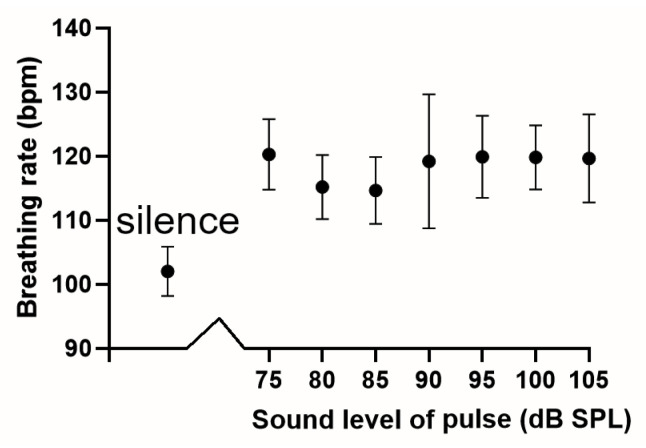
The breathing rate was measured in silence for eight guinea pigs after they had habituated to the recording chamber. It was measured again during each presentation of multiple series of startle pulses at different sound levels. There was a significant difference between the values for silence and those at 75 dB but no significant difference between the values of the different startle pulses. The error bars indicate the standard error of the mean.

## Data Availability

Original data can be accessed on request to the corresponding author. The data have not been summarized and organized. We can provide the corresponding original data as needed.

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
