# Peer review of "Breathing Rate as a Marker for Noise-Induced Stress in Guinea Pigs"

_brainsci, 2025, doi:10.3390/brainsci15111152_

Round 1
Reviewer 1 Report
Comments and Suggestions for Authors
The manuscript by Wallace and colleagues reports the detection of breathing-related abdominal movements (BRAMs) in guinea pigs—but not mice—in relation to acute loud-sound (acoustic startle) presentations. The findings are interesting, and the methods appear reproducible. I have several comments that could help improve the manuscript:
- Title scope
The title reads as an overstatement given that the experiments interrogate only auditory startle responses. Because stress can be elicited by many modalities, the title should be narrowed to the auditory domain. - Line 107: Sample size and animal details
Please specify how many mice were used. - Figures 4 and 5; GPIAS analysis
Figure 5 appears to present data from a “no-noise” condition. Because GPIAS was employed, the effect of the gap (gap vs. no-gap) on both startle magnitude and BRAMs should be analyzed explicitly. - Breathing metrics and time course
Beyond the reported increase in breathing rate after startle, consider analyzing additional features: inspiratory and expiratory durations, duty cycle, amplitude of abdominal displacement, variability (e.g., CV), and possibly latency to peak and recovery time back to baseline. Indicating how long the rate increase persists (e.g., half-life or time-to-baseline) would strengthen the physiological interpretation.
Author Response
"Please see the attachment."

Reviewer 2 Report
Comments and Suggestions for Authors
Thanks for the chance or learning from this manuscript. One of the key aspects of this study is the use of a motion tracking system with reflective markers in order to assess respiratory movements in animals. I believe that this is a valuable advancement, linking underlying physiological responses capturing stress reactivity. However, the need for improvement, particularly in the methodology, is imperative. As it stands, the manuscript fails to allow any replication.
- Punctuation, journal’s citation style and proper use of capital letters should be revised. Moreover, the wording of some sections are way too colloquial.
- Abstract: the methods section is way too short. Information on number of trials and detailed experimental conditions should be clarified.
- Still regarding abstract, please add how data was analyzed and be sure to include point estimates for statistics.
- The first paragraph claims that “recent studies in the rat have shown evidence that tinnitus can be produced”, but you cited two papers from the same authors. Do you have further evidence? This paragraph also contains different ideas, which would require independent paragraphs to ease the navigation by your audience.
- The same applies to the second paragraph of the introduction. It is dense and informative. However, it does not offer an in-depth explanation regarding why respiratory dynamics can be an accurate index of stress. Thus, the physiological rationale of the method is somewhat unexamined. For example, it could include very brief details about how stress-induced activation of the autonomic nervous system (e.g., due to sympathetic arousal and vagal mechanism) has a direct effect on respiratory rate.
- Please, add your objectives in an independent paragraph. Also pay close attention to the fact that it seems that you are presenting a mixture of methods/results in the introduction.
- The methods section needs much more detail. There are no explicit details on design (I can only assume it is a factorial design?). More strikingly, how was data analyzed? (see my comment about the abstract). I encourage authors to select a gold-standard reporting guideline according to the study’s design, and upload it along with the revised version of the text.
- You present R2 statistics, claiming it to be correlation. However, no information about data analysis was reported. R2 is the squared results of the correlation (r), so I assume our correlation was way below the reported number. P values are absent. I would remove figure 3.
- Your report on the main results comes nearly at the end of the section. Then you also state the use of non-parametric statistics. This should belong to the data analysis section, where you also need to report the appropriateness of the statistics given your sample size (that went down to 7, without clear explanations).
- Discussion: following the results, I was expecting to see an in-depth discussion about specific underlying physiological differences between species, which authors provided.
- The sniffing behavior is discussed, but I missed a more pronounced indication to inform scholars whether to distinguish sniffing as an exploration behavior or stress related.
- The following sentence is another example of the importance of clearly indicating your design in the appropriated section “The wide range of guinea pig respiratory rates observed both within an individual study such as ours and between different studies”.
- You finish your paper introducing the concept of anxiety. It is not clear if authors equal anxiety to stress and, if so, why change terminology?
- What are the conclusions?
Author Response
"Please see the attachment."

Round 2
Reviewer 2 Report
Comments and Suggestions for Authors
All the suggestions were addressed or explanations given (i.e., regulatory laws in the UK regarding limited time to conduct experiments). The physiological mechanisms were revamped and the discussion toned it down some of the claims that could not be inferred from the actual data. It seems that, indeed, this method is quite novel and I think this manuscript can be a basis for future studies.